# Field Study on the Prevalence of Ovine Footrot, Contagious Ovine Digital Dermatitis, and Their Associated Bacterial Species in Swedish Sheep Flocks

**DOI:** 10.3390/pathogens12101224

**Published:** 2023-10-08

**Authors:** Anna Rosander, Sophia Mourath, Ulrika König, Ann Nyman, Sara Frosth

**Affiliations:** 1Department of Biomedical Sciences and Veterinary Public Health, Faculty of Veterinary Medicine and Animal Science, Swedish University of Agricultural Sciences (SLU), P.O. Box 7036, 750 07 Uppsala, Sweden; anna.rosander@slu.se; 2District Veterinarians Roma, Visbyvägen 49, 622 54 Romakloster, Sweden; sophia.mourath@distriktsveterinarerna.se; 3Farm & Animal Health, Kungsängens Gård, 753 23 Uppsala, Sweden; ulrika.konig@gardochdjurhalsan.se; 4Department of Knowledge and Development, Växa, Ulls väg 29A, 751 05 Uppsala, Sweden; ann.nyman@vxa.se

**Keywords:** CODD, *Dichelobacter nodosus*, *Fusobacterium necrophorum*, *Treponema*, lesion scoring, photographs, swab samples, real-time PCR

## Abstract

Ovine footrot and contagious ovine digital dermatitis (CODD) cause lameness in sheep, affecting welfare and economics. Previous Swedish studies focused on individual slaughter lambs, leaving flock-wide prevalence less explored. This study examined the prevalence of footrot and CODD in Swedish sheep flocks, focusing on adult sheep. From 99 flocks, 297 swabs were analysed using real-time PCR for *Dichelobacter nodosus*, *Fusobacterium necrophorum*, and *Treponema* spp. Sampled feet were photographed and assessed using scoring systems for footrot and CODD. Results indicated footrot prevalences (footrot score ≥ 2) of 0.7% and 2.0% at the individual and flock levels, respectively, whereas there were no signs of CODD. The individual footrot prevalence was lower than that from a 2009 study but aligned with a 2020 study, both conducted on slaughter lambs. *Dichelobacter nodosus*, *F. necrophorum*, and *Treponema* spp. were found in 5.7%, 1.3%, and 65.0% of sheep, and in 9.1%, 3.0%, and 82.8% of flocks, respectively. Compared to the 2020 study, there was a notable decrease in *F. necrophorum* and *Treponema* spp., while *D. nodosus* was consistent. In conclusion, the findings show a low prevalence of footrot, CODD, *D. nodosus*, and *F. necrophorum* in Swedish sheep flocks. Continuous surveillance and owner education are important to maintain this favourable status.

## 1. Introduction

Ovine footrot and contagious ovine digital dermatitis (CODD) are infectious foot diseases that can lead to lameness in sheep. These diseases not only cause extensive damage to the feet of affected animals but also inflict suffering and result in economic losses for sheep farmers. *Dichelobacter nodosus* has long been identified as the aetiological agent for ovine footrot [1], while the exact cause of CODD remains undetermined. The initial description of CODD was documented in 1997 in the United Kingdom [2]. Subsequent studies, including those by Sayers et al. [3] and Sullivan et al. [4], have associated CODD with species of the *Treponema* genus, particularly *Treponema phagedenis*, *Treponema pedis*, and *Treponema medium.* Recently, Duncan et al. [5] conducted an experimental study suggesting that footrot and CODD might be different stages of the same disease. However, CODD has traditionally been considered distinct from footrot due to various factors, including its specific presentation and response to treatments [5]. Nonetheless, the involvement of *D. nodosus*, *Fusobacterium necrophorum*, *T. phagedenis*, *T. pedis*, and *T. medium* are recognised as important in disease development [6]. The severity of footrot is influenced by the virulence of the infecting *D. nodosus* strain [7]. Real-time PCR assays are available that can distinguish between benign and virulent *D. nodosus* variants [8,9], highlighting the importance of the virulence testing of *D. nodosus* isolates. Additionally, *F. necrophorum* has been identified as a contributing factor in exacerbating footrot severity [1,10]. Notably, differences in virulence exist between the two *F. necrophorum* subspecies. In particular, the *F. necrophorum* subsp. *necrophorum* has been found to be more virulent [11], underscoring the need to differentiate between the subspecies.

In Sweden, footrot was first diagnosed in 2004 [12], followed by the identification of CODD in 2019 [13]. A voluntary control program for footrot, known as Klövkontrollen, was initiated in 2009 and expanded to include CODD in 2019. By the end of 2022, 4.7% of Swedish sheep flocks (396 out of 8488) were enrolled in the program [14]. Previous prevalence studies based on feet from slaughter lambs were conducted in 2009 [15] and 2020 [16]. These studies reported individual-level prevalence rates of 5.8% and 1.8% for footrot, respectively. In the study by Rosander et al. [16], suspected CODD was identified in one lamb (0.2%), while *D. nodosus*, *F. necrophorum*, and *Treponema* spp. were found in 6.1%, 7.6%, and 90.6% of the lambs, respectively. However, these prevalences may be underestimated because lame animals are not sent for slaughter. Furthermore, prevalence numbers were obtained only at the individual level, while footrot and CODD are typical flock diagnoses due to their contagious nature [1,17]. 

Therefore, the primary objective of this study was to address the existing knowledge gap regarding the prevalence of footrot and CODD in Swedish sheep flocks and contribute to the understanding and control of these diseases through a field study investigating adult sheep.

## 2. Materials and Methods

### 2.1. Study Farms and Period

In April 2022, a request for voluntary participation in the study was sent by e-mail to all sheep farm owners who reported that they had at least ten animals and had provided an e-mail address with the Swedish Board of Agriculture in December 2021; there was a total of 5593 farms. Study farms were chosen on a first-come, first-served basis to achieve a randomised selection, and 125 farms were recruited within 24 h. The number of study farms needed was calculated using the sample size Equation [18]
n = (Z^2^ × P (1 − P))/d^2^
and was found to be 75 farms, but extra farms were signed up to compensate for possible dropouts, loss of samples, and diagnostic sensitivity. In the calculation, the prevalence (P) was estimated to be 5%, the precision (d) was set to 5%, and a confidence level of 95% was used. The number of farms that completed the study was 99. Dropouts occurred due to reasons such as the animals being sold, culling due to parasitic infestations, the owner’s illness, or a lack of time. 

The study was conducted in accordance with Swedish animal health regulations and met the conditions for exemption from the requirement for ethical permission. All animals were privately owned, and no compensation was paid to the sheep farmers. Participating sheep farmers signed a consent form where they confirmed that they allowed their animals to participate in the study and that they understood that they could withdraw their participation at any time and without cause. The sampling did not cause pain (no skin penetration), stress, or suffering for the animals. 

The study period spanned September to October (2022), aligning with the annual foot inspection period of the control program and previous prevalence studies [15,16].

### 2.2. Sampling, Photography, and Lesion Assessment of Sheep Feet

Sampling was performed by the sheep farmers after detailed written and video-based instructions. The sheep farmers were asked to sample three of their animals that were at least one year of age and to prioritise animals with visibly damaged feet and/or lameness. In the absence of damaged feet or lameness, they were asked to sample the left forefoot of three randomly selected animals that were at least one year old. In cases where the left forefoot was heavily encrusted with manure, the sheep farmers were instructed to choose another animal. In Sweden, the average flock size (ewes and rams) is approximately 32 [19]. The number of animals sampled per flock was chosen based on practical considerations. Given the contagious nature of both footrot and CODD, they are typically diagnosed at the flock level [1,17]. To enhance the likelihood of detecting these conditions, if present in the flock, animal owners were directed to preferentially select individuals displaying visible foot damage and/or lameness.

The samples were taken with Eswabs (Copan Innovation Ltd., Brescia, Italy) in the interdigital skin (one per animal) and sent by regular mail to the Department of Biomedical Sciences and Veterinary Public Health at the Swedish University of Agricultural Sciences (SLU). In addition to the sampling, the sheep farmers were also requested to photograph the sampled feet by taking one photograph of the interdigital skin and one photograph from the front of the foot showing the coronary band for each sampled animal. The photographs were sent in by e-mail. Lesion assessment was performed as previously described by Rosander et al. [16] using the footrot scoring system by Stewart and Claxton [20], with the definition of footrot as the presence of at least one foot with a footrot score ≥ 2, and the CODD grading system by Angell et al. [21]. However, photographs were used instead of examining the feet directly. 

### 2.3. Bacterial DNA Extraction and Real-Time PCR

Bacterial DNA was extracted from the swab samples (*n =* 297) as previously described [22]. In short, swab samples were pelleted and pre-treated with G2-lysis buffer and proteinase K before extraction based on magnetic bead separation using an EZ1 Advanced XL (Qiagen, Hilden, Germany). The elution volume used was 100 µL, and the DNA eluates were stored at −20 °C awaiting real-time PCR analysis. 

The DNA eluates were analysed using real-time PCR for the detection of *D. nodosus* via a 16S rRNA assay, and positive samples were analysed using an *aprV2/B2* PCR for virulence determination, as both previously described [9]. All DNA eluates were also analysed for *F. necrophorum* and *Treponema* spp., as described by Rosander et al. [16], with primers and probes originating from Jensen et al. [23] and Strub et al. [24], respectively. The *Treponema* spp. PCR primers used did not enable differentiation to species level. In addition, a quantitative (q) PCR targeting *T. phagedenis*, *T. pedis*, *T. medium*, and ‘*Treponema vincentii*’ was deployed on all DNA eluates [25]. Amplification was carried out on a CFX Opus 96 Real-Time PCR Instrument (Bio-Rad Laboratories Inc., Hercules, CA, USA) and analysed with the CFX Maestro Software version 2.3 (Bio-Rad Laboratories Inc.) with default settings. The threshold for positive samples used was quantification cycle (Cq) < 40. DNase- and RNase-free water (Sigma-Aldrich, St Louis, MO, USA) was used as a negative control in all PCR runs, and all runs included positive controls. In the *D. nodosus* assays, strains AN363/05 (*aprV2*-positive) and AN484/05 (*aprB2*-positive) were used. For the detection of *F. necrophorum*, strains *F. necrophorum* subsp. *necrophorum* CCUG 9994T and *F. necrophorum* subsp. *funduliforme* CCUG 42162T (Culture Collection, University of Gothenburg, Sweden) were used. *Treponema pedis* DSM 18691 (Leibniz Institute DSMZ-German Collection of Microorganisms and Cell Cultures GmbH, Braunschweig, Germany) was used in the *Treponema* spp. assay. A linearised plasmid containing the three target sequences for *T. phagedenis*, *T. pedis*, and *T. medium/’T. vincentii’* was used for the *Treponema* species-specific assay described by Frosth et al. [25]. 

### 2.4. Statistical Analysis

The Clopper–Pearson confidence interval formula was used to determine the 95% confidence interval (CI) for binomial data. The two proportion z test was used for several comparisons: to contrast the prevalences in this study with findings from the nation’s two previous prevalence studies [15,16], to evaluate the difference between the number of flocks affiliated with the footrot and CODD control program and the total number of flocks, and to assess the association between the presence of *D. nodosus* and affiliation status.

## 3. Results

### 3.1. Study Farms

The study farms were geographically distributed across Sweden (Figure 1). Flock sizes varied ranging from 10 to 200 ewes and rams (Figure 2). The sheep ages spanned from 1 to 13 years, with an average age of 4 years. Among the breeds, crossbred sheep were the most frequently sampled at 23.9%, followed by Gotland Pelt sheep at 20.2%. Thirteen of the study farms (13.1%) were affiliated with the national control program for footrot and CODD. In contrast, only 4.7% [14] of all Swedish sheep farms held such an affiliation at that time. The difference in affiliation rates between the study farms and the national average was statistically significant (*p* < 0.01). 

### 3.2. Assessment of Photographs

In total, photographs from 297 animals and 99 sheep flocks were assessed for footrot and CODD. However, 17 animals were excluded from the assessment either due to missing photographs, because the sheep farmer failed to submit them (*n* = 6), or because the provided photographs were of insufficient quality (*n* = 11). Consequently, the footrot and CODD status of these animals were considered as unknown. 

In total, 2 animals were assessed to have a footrot score of 2 (0.7%; CI 0.1–2.4%) (Figure 3), 5 animals had a footrot score of 1 (1.7%; CI 0.5–3.9%) (Figure 4), 273 animals had a footrot score of 0 (91.9%; CI 88.2–94.8%) (Figure 5), and the remaining 17 (5.7%; CI 3.4–9.0%) were of unknown status (Table 1). The individual footrot prevalence (footrot score ≥ 2) of 0.7% was significantly lower than the prevalence reported in the 2009 study (5.8%, *p* < 0.01) by König et al. [15]. However, there was no significant difference in footrot prevalence (footrot score ≥ 2) compared to the 2020 study (1.8%, *p* = 0.20) [16].

At the flock level, two flocks had animals with a footrot score of 2 (2.0%; CI 0.2–7.1%), three flocks had animals with a footrot score of 1 (3.0%; CI 0.6–8.6%), ninety flocks had animals with a footrot score of 0 (90.1%; CI 83.4–95.8%), and the remaining four (4.0%; CI 1.1–10.0) were of unknown status (Table 2). No animals were assessed to have a footrot score > 2 or visual signs of CODD.

### 3.3. Real-Time PCR Analysis of Swab Samples

*Dichelobacter nodosus* was detected in 17 out of 297 (5.7%; CI 3.4–9.0%) animals (Table 1). The prevalence at the individual level did not significantly differ from that of the most recent prevalence study (6.1%; *p* = 0.82) [16]. At the flock level, *D. nodosus* was detected in 9 out of 99 (9.1%; CI 4.2–16.6%) sheep flocks (Table 2). Of the nine flocks with *D. nodosus*, one flock was affiliated with the control program, while the remaining eight were not. There was no statistically significant difference between the presence of *D. nodosus* and affiliation to the control program (*p* = 0.85). *Dichelobacter nodosus* was detected in both animals with a footrot score of 2, and in three out of five animals with a footrot score of 1. In addition, *D. nodosus* was detected in 12 animals with a footrot score of 0 and no visual signs of CODD. All 17 *D. nodosus* were shown to be benign by the real-time PCR detecting the *aprV2/B2* genes. 

*Fusobacterium necrophorum* subsp. *funduliforme* was detected in 4 out of 297 (1.3%; CI 0.4–3.4%) animals (Table 1) and in 3 out of 99 (3.0%; CI 0.6–8.6%) sheep flocks (Table 2). Of these, the bacterium occurred simultaneously with *D. nodosus* and a footrot score of 1 in one animal. Of the other positive samples, two occurred together with *Treponema* spp. The subspecies *F. necrophorum* subsp. *necrophorum* was not detected in any sample. The individual-level prevalence of *F. necrophorum* (1.3%) was significantly lower than that reported in the study by Rosander et al. [16] (7.6%, *p* < 0.01).

*Treponema* spp. was detected in 193 out of 297 (65.0%; CI 59.3–70.4%) animals (Table 1). The prevalence at the individual level differed significantly from that of the most recent prevalence study (90.6%; *p* < 0.01) [16]. At the flock level, *Treponema* spp. was detected in 82 out of 99 (82.8%; CI 73.9–89.7%) sheep flocks (Table 2). *Treponema* spp. was detected in 15 out of 17 (88.2%) animals that were positive for *D. nodosus* and in 9 out of 9 (100%) sheep flocks where *D. nodosus* was present. *Treponema* spp. was detected in seven out of seven (100%) animals with footrot scores of 1 or 2, and *D. nodosus*, *Fusobacterium necrophorum* subsp. *funduliforme*, and *Treponema* spp. were found in combination in one animal with a footrot score of 1. None of the CODD-associated *Treponema* species (*T. phagedenis*, *T. pedis*, and *T. medium*) were detected in any of the samples. 

## 4. Discussion

Previous Swedish prevalence studies on footrot and/or CODD have examined slaughter lambs [15,16]. One disadvantage with these studies is that animals sent for slaughter are typically young and not expected to have any diseases. As a result, the sample group is not entirely representative of the overall sheep population. In this study, we investigated the prevalence of footrot and CODD in adult sheep (≥1 year) through a field study, which, for the first time, also enabled results to be obtained at the flock level. While our choice of the number of animals sampled per flock was largely dictated by practical considerations, this approach might not encompass the entire range of footrot and CODD prevalence, especially in larger flocks. To mitigate this, we directed animal owners to prioritise animals displaying visible foot damage or lameness, aiming to enhance the probability of detecting these conditions if present, regardless of the smaller sample size. The robustness of our prevalence estimation at the individual level is underscored by the fact that we sampled a total of 297 animals, whereas our calculations indicated a requirement of merely 75.

The sheep farms studied were geographically spread across the country and reflected the sheep density relatively well, except for the county of Gotland. Despite having the highest sheep density, Gotland only had one participating farm (Figure 1). This limited participation from Gotland might be attributed to the few cases of footrot reported there, a trend that aligns with the latest prevalence study on slaughter lambs [16]. The flock size distribution in our study was generally representative of the country (Figure 2). However, the study had a higher proportion of farms with 10–24 and 25–49 animals, likely because we excluded the smallest farms (1–9 animals). These small farms, though numerous, comprise just 5.0% of the total sheep population [19]. Notably, our study revealed a higher affiliation rate with the footrot and CODD control program compared to the national average, potentially suggesting heightened disease awareness among the participating farmers. This heightened awareness could influence the interpretation of the results, as farms with greater awareness might have different management practices. However, there was no observed association between the presence of *D. nodosus* and affiliation status. 

In this study, the prevalence of footrot (defined in Sweden as the presence of at least one foot with a footrot score ≥ 2, according to the scoring system devised by Stewart and Claxton [20]) at the flock level was 2.0% among Swedish sheep flocks. At the individual level, footrot prevalence (footrot score ≥ 2) was 0.7%, which is significantly lower than the prevalence reported in the 2009 study by König et al. [15], but not significantly different compared to the 2020 study [16]. Contagious ovine digital dermatitis was not detected in this study at either the flock or individual level. This finding was not surprising given that the disease has only been identified in Sweden on two prior occasions, after which all animals in the affected flocks were slaughtered [13]. The rigorous measures taken might have effectively halted the disease’s spread, potentially explaining the absence of CODD observations in this study. Although the 2009 study did not investigate CODD, the 2020 study identified only one single lamb with suspected CODD. The relatively short time span between the 2020 study and the present study facilitates a more direct comparison, suggesting no statistical disparity between footrot and CODD prevalence in adult sheep and lambs in Sweden. However, significantly fewer animals were assessed to have a footrot score of 1, and thus significantly more animals had a footrot score of 0 in this study compared to the study conducted in 2020 [16]. This difference could possibly be explained by the difficulty of assessing feet from photographs. However, within the footrot and CODD program, the submission of photographs for assessment has been a simple, efficient, and rapid method for investigating lameness. Field veterinarians have the option to forward these photographs to the program coordinator when faced with unclear cases. Moreover, these photographs are used in the annual calibration process for evaluating the expertise of veterinarians trained in the program. However, the quality of these photographs is essential. As demonstrated in this study, it is crucial to provide clear and detailed instructions to those capturing the photographs to ensure consistency and accuracy.

The prevalence of *D. nodosus* was 9.1% among Swedish sheep flocks and 5.7% in individual sheep. The prevalence at the individual level showed neither an increase nor a decrease when compared to the most recent prevalence study [16]. Since this study represents the first investigation of *D. nodosus* at the flock level in Sweden, it is not possible to determine whether the prevalence has increased or decreased in the country over time. Recently, the prevalence of *D. nodosus* in German sheep flocks was determined to be 71% [26], and in comparison to this, the occurrence in Swedish sheep flocks is low. The low occurrence of the bacterium in Sweden compared to other countries may be attributed to several factors, including a generally lower stocking density and climatic conditions. Similar to previous Swedish studies [16,27], no virulent strains of *D. nodosus* could be detected, which differs from, for example, Germany, Switzerland, and the United Kingdom, where virulent strains are more commonly found than benign strains [26,28,29,30]. Virulent strains have, however, been detected in Sweden, but not to any great extent [9,31]. 

Contrary to its previous classification as ubiquitous, a recent study has indicated that *F. necrophorum* is host-bound, present in the faeces of a limited number of individuals within a flock, and predominantly found in feet affected by footrot [10]. In the present study, the prevalence of *F. necrophorum* among Swedish sheep flocks was 3.0%, while the individual-level prevalence was 1.3%, with only the lesser virulent subspecies *funduliforme* being detected. The observed difference in prevalence from the 7.6% reported by Rosander et al. [16] could potentially be attributed to variations in *F. necrophorum* between lambs and adult sheep.

The prevalence of *Treponema* spp. among Swedish sheep flocks was 83%, whereas the individual-level prevalence was 65%. Although the prevalence at the individual level has decreased since the 2020 study [16], it remains at a high level, and the significance of the high prevalence of *Treponema* spp. in Swedish lambs and adult sheep is unknown. However, none of the three CODD-associated *Treponema* species, *T. phagedenis*, *T. pedis*, and *T. medium,* were detected in any of the flocks or individuals. *Treponema* species, which have not been associated with bovine digital dermatitis, have been detected in the gastrointestinal tract of cattle and are thought to be present in the barn environment [32]. Therefore, it is possible that the *Treponema* bacteria identified in this study might naturally exist in sheep without inducing disease.

The observed low prevalence of footrot, CODD, *D. nodosus*, and *F. necrophorum* in this study, when combined with the findings from previous studies, suggests that Sweden is in a favourable situation with respect to these diseases. However, of particular concern is the recently abolished quarantine requirement for the movement of sheep between EU countries, as stipulated by EU Regulation 2020/688. In light of this new EU legislation, it is imperative for animal owners to be more informed and proactive to mitigate the risk of introducing these diseases into Sweden.

To conclude, this study indicates a low prevalence of footrot, CODD, *D. nodosus*, and *F. necrophorum* in Swedish sheep flocks. The high prevalence of *Treponema* spp. is noteworthy. While the three CODD-associated *Treponema* species were not detected, the significance of this high prevalence remains to be deciphered. The use of a field study approach, encompassing a broader age range and geographical distribution, has allowed for a more comprehensive understanding of the disease distribution in the country. However, with relaxed EU quarantine measures, there is an elevated risk of disease introduction. Continuous vigilance and education among animal owners are essential to maintain this favourable status.

## Figures and Tables

**Figure 1 pathogens-12-01224-f001:**
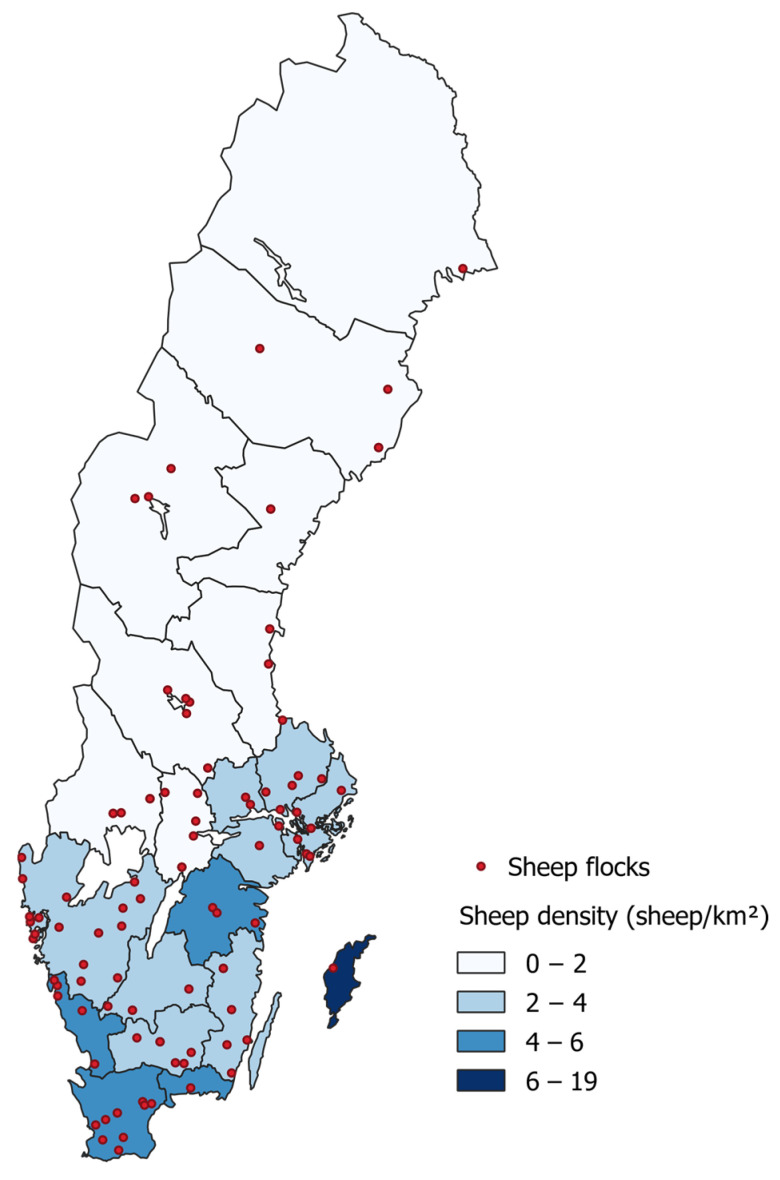
Localisation of study sheep farms relative to county-level sheep density. Figure created in QGIS 3.26.3−Odense (Free Software Foundation Inc., Boston, MA, USA) using sheep data from the Swedish Board of Agriculture [19], and map and area data from Statistics Sweden https://scb.se/ (accessed on 25 March 2023).

**Figure 2 pathogens-12-01224-f002:**
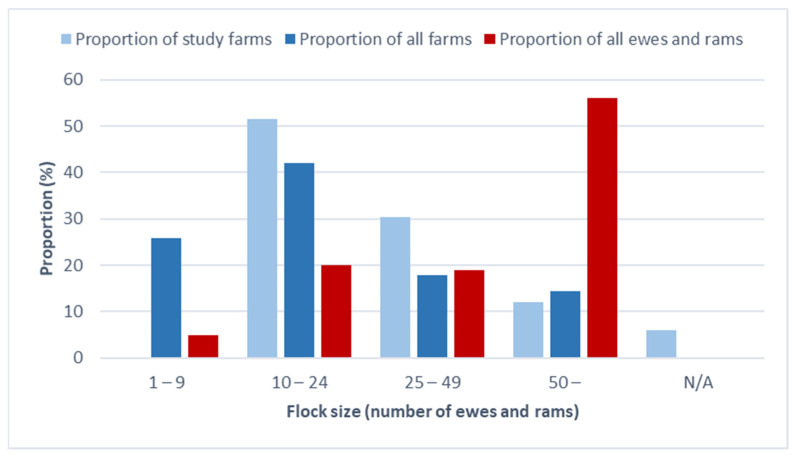
Distribution of flock sizes (ewes and rams) in study sheep farms. Data on study farms provided by the animal owners at the time of sampling and data on all farms and all ewes and rams in Sweden is from the Swedish Board of Agriculture [19]. N/A = Not available.

**Figure 3 pathogens-12-01224-f003:**
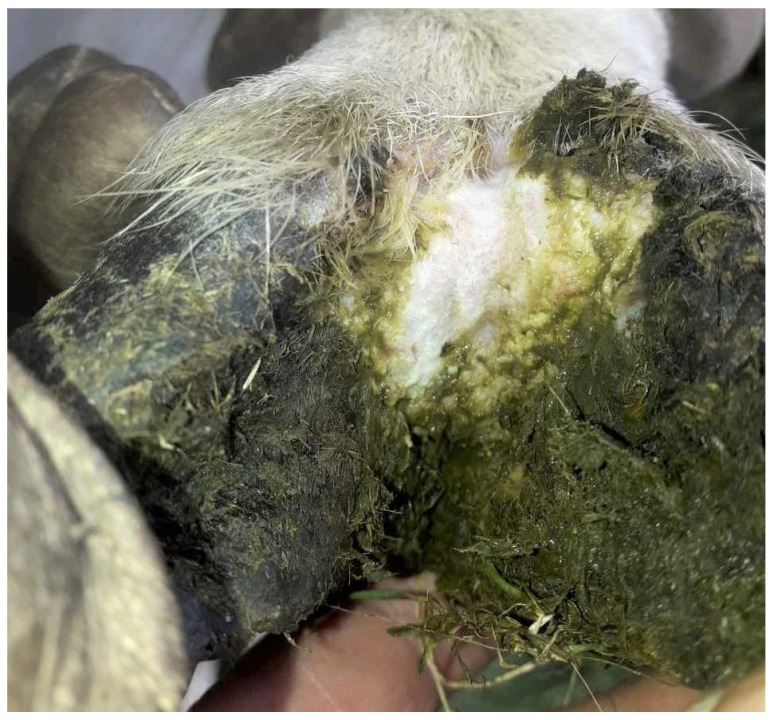
Photograph of a sheep foot displaying characteristics of a footrot score of 2 as defined by Stewart and Claxton [20]: necrotising inflammation of the interdigital skin and skin horn junction. This photo is representative of the typical footrot score 2 presentations observed in the current study.

**Figure 4 pathogens-12-01224-f004:**
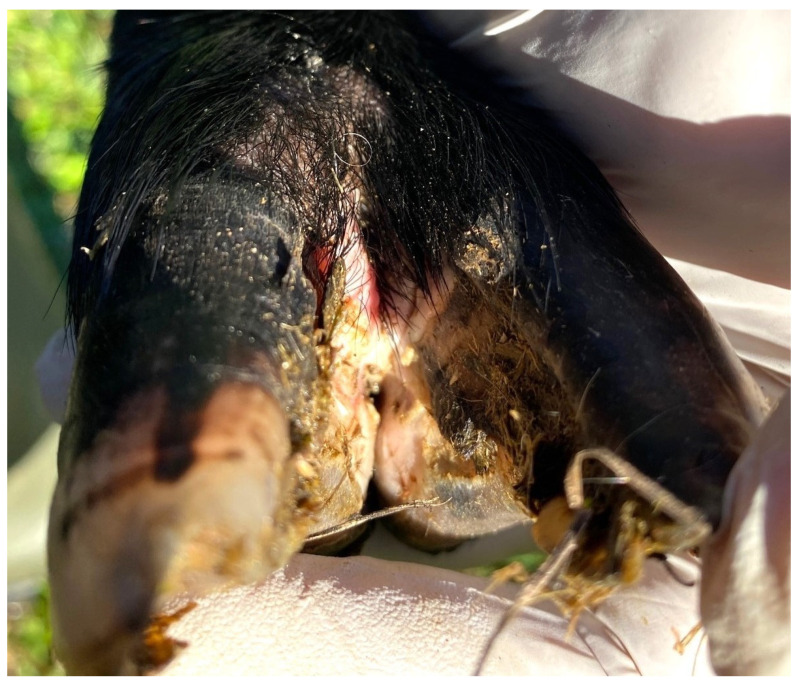
Photograph of a sheep foot displaying characteristics of a footrot score of 1 as defined by Stewart and Claxton [20]: inflammation of the interdigital skin. This photo is representative of the typical footrot score 1 presentations observed in the current study.

**Figure 5 pathogens-12-01224-f005:**
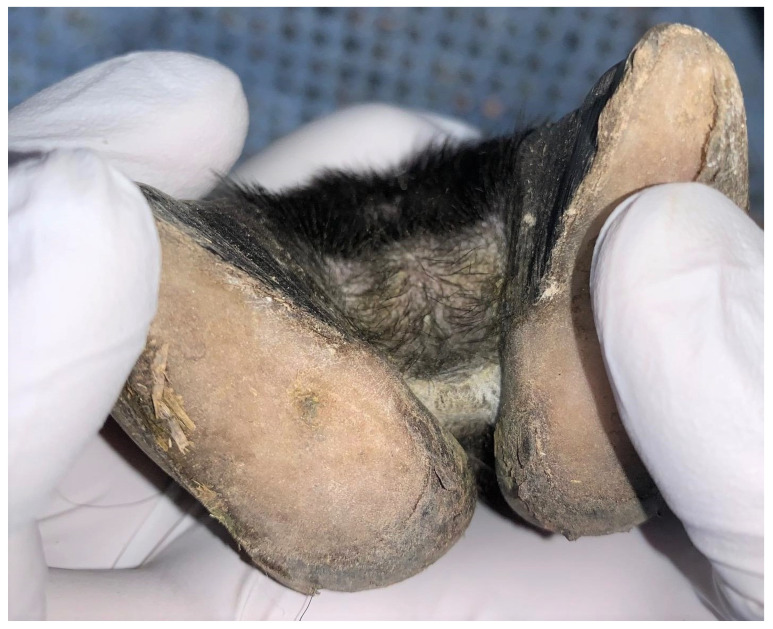
Photograph of a sheep foot displaying characteristics of a footrot score of 0 as defined by Stewart and Claxton [20]: healthy foot. This photo is representative of the typical footrot score 0 presentations observed in the current study.

**Table 1 pathogens-12-01224-t001:** Distribution of *Dichelobacter nodosus*, *Fusobacterium necrophorum*, *Treponema* spp., and footrot scores in Swedish sheep.

	Footrot Score 0 (%)	Footrot Score 1 (%)	Footrot Score 2 (%)	Footrot Score Unknown (%)	Total
Number of sheep	273 (91.9)	5 (1.7)	2 (0.7)	17 (5.7)	297
Number of sheep with *D. nodosus*	12 (4.0)	3 (1.0)	2 (0.7)	0 (0)	17
Number of sheep with *F. necrophorum*	3 (1.0)	1 (0.3)	0 (0)	0 (0)	4
Number of sheep with *Treponema* spp.	178 (60.0)	5 (1.7)	2 (0.7)	8 (2.7)	193

**Table 2 pathogens-12-01224-t002:** Distribution of *Dichelobacter nodosus*, *Fusobacterium necrophorum*, *Treponema* spp., and footrot scores in Swedish sheep flocks.

	Footrot Score 0 (%)	Footrot Score 1 (%)	Footrot Score 2 (%)	Footrot Score Unknown (%)	Total
Number of sheep flocks	90 (90.1)	3 (3.0)	2 (2.0)	4 (4.0)	99
Number of sheep flocks with *D. nodosus*	6 (6.1)	1 (1.0)	2 (2.0)	0 (0)	9
Number of sheep flocks with *F. necrophorum*	2 (2.0)	1 (1.0)	0 (0)	0 (0)	3
Number of sheep flocks with *Treponema* spp.	75 (75.8)	3 (3.0)	2 (2.0)	2 (2.0)	82

## Data Availability

The data presented in this study are available on request from the corresponding author.

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
