# Peer review of "Field Study on the Prevalence of Ovine Footrot, Contagious Ovine Digital Dermatitis, and Their Associated Bacterial Species in Swedish Sheep Flocks"

_pathogens, 2023, doi:10.3390/pathogens12101224_

Round 1

Reviewer 1 Report

The authors conducted a comprehensive investigation into the prevalence of ovine footrot and contagious ovine digital dermatitis within Swedish sheep flocks, with a particular emphasis on adult sheep. To accomplish their objectives, they employed photography methods and real-time PCR assays targeting Dichelobacter nodosus, Fusobacterium necrophorum, and Treponema spp.

While the study utilized fundamental research tools, it is noteworthy that the chosen methodology aligns with established practices within the field. Nevertheless, I would like to suggest the inclusion of photographic documentation depicting sheep feet, encompassing both typical and various atypical conditions. Incorporating these visuals as supplementary data or within the main text would undoubtedly enhance the report's comprehensibility, providing readers with valuable visual insights.

Regarding the real-time PCR analysis of Treponema spp., is it feasible to pursue further characterization through sequencing of the PCR product? This additional step would facilitate the precise identification of Treponema species, offering a deeper understanding of the microbial composition associated with ovine digital dermatitis. Such molecular profiling can contribute substantially to the etiological understanding and potential therapeutic strategies for this condition.

Author Response

Comments and Suggestions for Authors

The authors conducted a comprehensive investigation into the prevalence of ovine footrot and contagious ovine digital dermatitis within Swedish sheep flocks, with a particular emphasis on adult sheep. To accomplish their objectives, they employed photography methods and real-time PCR assays targeting Dichelobacter nodosus, Fusobacterium necrophorum, and Treponema spp.

While the study utilized fundamental research tools, it is noteworthy that the chosen methodology aligns with established practices within the field. Nevertheless, I would like to suggest the inclusion of photographic documentation depicting sheep feet, encompassing both typical and various atypical conditions. Incorporating these visuals as supplementary data or within the main text would undoubtedly enhance the report's comprehensibility, providing readers with valuable visual insights.

Answer: Thank you very much for this valuable comment. We have now included photographs in the main text that are typical of footrot scores 0, 1, and 2, respectively, and represent our study's findings. We have also added a few sentences about the current use of photographs for assessing lameness in the Swedish footrot and CODD control program to the discussion as follows: “However, within the footrot and CODD program, submission of photographs for assessment has been a simple, efficient, and rapid method for investigating lameness. Field veterinarians have the option to forward these photographs to the program coordinator when faced with unclear cases. Moreover, these photographs are used in the annual calibration process for evaluating the expertise of veterinarians trained in the program. However, the quality of these photographs is essential. As demonstrated in this study, it is crucial to provide clear and detailed instructions to those capturing the photographs to ensure consistency and accuracy.”

Regarding the real-time PCR analysis of Treponema spp., is it feasible to pursue further characterization through sequencing of the PCR product? This additional step would facilitate the precise identification of Treponema species, offering a deeper understanding of the microbial composition associated with ovine digital dermatitis. Such molecular profiling can contribute substantially to the etiological understanding and potential therapeutic strategies for this condition.

Answer: Thank you for your suggestion. However, this specific qPCR product is only 128 bp long and is not long enough to discriminate between the different Treponema species.

Author Response

Comments and Suggestions for Authors

This study reports prevalence of footrot and CODD and associated bacterial pathogens in Swedish sheep flocks. Previous studies have not reported flock level prevalence therefore this study provides novel information that is useful for ongoing footrot control in Sweden. Overall, the paper is clear and well written. My main query would be regarding how likely it is to detect presence of bacterial pathogens in flocks using 3 swabs per flock, and further information on this point is required for interpretation of the results. The discussion lacks depth in places. Please see below for more detailed comments.

Introduction

Line 43 – Whilst in the study reported by Duncan et al. cases of CODD were observed to occur following footrot, I would question whether there is sufficient evidence to support the statement that footrot and CODD are stages of the same disease.

Answer: Thank you for your insightful comment. In the study conducted by Duncan et al., based on their experimental findings, they suggest that the three disease states (ID, footrot & CODD) might be considered as stages in a consistent spectrum of ovine infectious foot disease. Duncan et al. also recognise the traditional distinction between footrot and CODD but highlight that their experimental findings provide a perspective that sees them as possibly connected. We understand the complexities and nuances of this interpretation. To ensure clarity, we have changed the text to the following “Recently, Duncan et al. [5] conducted an experimental study suggesting that footrot and CODD might be different stages of the same disease. However, CODD has traditionally been considered distinct from footrot due to various factors, including its specific presentation and response to treatments [5]. Nonetheless, the involvement of D. nodosus, Fusobacterium necrophorum, T. phagedenis, T. pedis, and T. medium are recognised as important in disease development [6].”

During the paper, virulent and benign strains of D. nodosus are described, as are subspecies of F. necrophorum. It would be helpful to provide some background/context for these in the introduction including findings from previous Swedish studies. Otherwise, it is not clear to the reader why it is useful to look specifically at different strains/subspecies.

Answer: Thank you for this suggestion. We have added the following paragraph to the introduction “The severity of footrot is influenced by the virulence of the infecting D. nodosus strain [7]. Real-time PCR assays are available that can distinguish between benign and virulent D. nodosus variants [8,9], highlighting the importance of virulence testing of D. nodosus isolates. Additionally, F. necrophorum has been identified as a contributing factor in exacerbating footrot severity [1,10]. Notably, differences in virulence exist between the two F. necrophorum subspecies. In particular, the F. necrophorum subsp. necrophorum has been found to be more virulent [11], underscoring the need to differentiate between the subspecies.”

Materials and Methods

Line 91 – some justification for the use of 3 sheep per flock is needed here. A sample size calculation is provided for the number of flocks, but not for number of sheep per flock. An indication of how likely it would be to detect a pathogen (given a likely prevalence) in flocks of different sizes using samples from 3 sheep is necessary for interpretation of the results. Or alternatively, how many samples would be required to support the hypothesis that a pathogen is absent from a flock?

Answer: We appreciate the reviewer's insight regarding the intra-flock sampling. The decision to sample 3 sheep per flock was due to practical reasons. The following information has been added to materials and methods: “In Sweden, the average flock size (ewes and rams) is approximately 32 [19]. The number of animals sampled per flock was chosen based on practical considerations. Given the contagious nature of both footrot and CODD, they are typically diagnosed at the flock level [1,17]. To enhance the likelihood of detecting these conditions, if present in the flock, animal owners were directed to preferentially select individuals displaying visible foot damage and/or lameness” and the following has been added to the discussion section “While our choice of the number of animals sampled per flock was largely dictated by practical considerations, this approach might not encompass the entire range of footrot and CODD prevalence, especially in larger flocks. To mitigate this, we directed animal owners to prioritise animals displaying visible foot damage or lameness, aiming to enhance the probability of detecting these conditions if present, regardless of the smaller sample size. The robustness of our prevalence estimation at the individual level is underscored by the fact that we sampled a total of 297 animals, whereas our calculations indicated a requirement of merely 75.”.

Results

Line 182 – the method for determining association between presence of D. nodosus and affiliation to the control program is not described in the methods.

Answer: Thank you for noticing this. The following sentence has been added to materials and methods: “The two proportion z test was used for several comparisons: to contrast the prevalences in this study with findings from the nation's two previous prevalence studies [15,16], to evaluate the difference between the number of flocks affiliated with the footrot and CODD control program and the total number of flocks, and to assess the association between the presence of D. nodosus and affiliation status.”

Line 197 – the second part of this sentence might be clearer if the order was reversed i.e. ‘D. nodosus, Fusobacterium necrophorum subsp. funduliforme and Treponema spp. were found in combination in one animal with footrot score 1.’

Answer: Thank you for your suggestion. The sentence has been reversed accordingly.

Discussion

Line 208 – do the authors have any suggestions for why participation was low from farmers in Gotland, or whether this is likely to have impacted results?

Answer: Thank you for suggesting this. The sentence “This limited participation from Gotland might be attributed to the few cases of footrot reported there, a trend that aligns with the latest prevalence study on slaughter lambs [11].” has been added.

Line 216 – this result regarding difference in affiliation to the control program between the study flocks and all Swedish sheep flocks needs to be presented in results (line 144).

Answer: Thank you for your suggestion. The following sentence has been added to line 144: “In contrast, only 4.7% [14] of all Swedish sheep farms held such an affiliation at that time. The difference in affiliation rates between the study farms and the national average was statistically significant (p < 0.01).”

Line 217 – what could the implication of this heightened awareness be for interpretation of the study results?

Answer: Thank you for your question. This sentence has been expanded to “Notably, our study revealed a higher affiliation rate with the footrot and CODD control program compared to the national average, potentially suggesting heightened disease awareness among the participating farmers. This heightened awareness could influence the interpretation of the results, as farms with greater awareness might have different management practices. However, there was no observed association between the presence of D. nodosus and affiliation status.”

Line 218 – the definition of footrot used should be provided in the methods as well at line 105.

Answer: Thank you, this information has been added as suggested.

Lines 222-224, 240, 257, 262 – please provide results of statistical tests in results rather than discussion.

Answer: Thank you for this suggestion. The results from the statistical tests have been moved from the discussion to the results.